# The Prevalence of MRI-Defined Sacroiliitis and Classification of Spondyloarthritis in Patients with Acute Anterior Uveitis: A Longitudinal Single-Centre Cohort Study

**DOI:** 10.3390/diagnostics12010161

**Published:** 2022-01-11

**Authors:** Kristyna Bubova, Lenka Hasikova, Katerina Mintalova, Monika Gregova, Petr Kasalicky, Aneta Klimova, Michaela Brichova, Petra Svozilkova, Jarmila Heissigerova, Jiri Vencovsky, Karel Pavelka, Ladislav Senolt

**Affiliations:** 1Institute of Rheumatology, 12850 Prague, Czech Republic; hasikova@revma.cz (L.H.); mintalova@revma.cz (K.M.); gregova@revma.cz (M.G.); vencovsky@revma.cz (J.V.); pavelka@revma.cz (K.P.); senolt@revma.cz (L.S.); 2Department of Rheumatology, 1st Faculty of Medicine, Charles University, 12850 Prague, Czech Republic; 3Affidea Praha s. r. o., 14800 Prague, Czech Republic; kasalickyp@affidea-praha.cz; 4Department of Ophthalmology, 1st Faculty of Medicine, Charles University and General University Hospital in Prague, 12808 Prague, Czech Republic; aneta.klimova@volny.cz (A.K.); Michaela.Brichova@vfn.cz (M.B.); Petra.Svozilkova@lf1.cuni.cz (P.S.); jarmila.heissigerova@vfn.cz (J.H.)

**Keywords:** spondyloarthritis, axial spondyloarthritis, peripheral spondyloarthritis, uveitis, sacroiliitis, magnetic resonance imaging

## Abstract

Background: Acute anterior uveitis (AAU) is a relatively common extra-musculoskeletal manifestation of axial spondyloarthritis (axSpA); however, data on the prevalence of active sacroiliitis in patients with AAU are limited. Methods: 102 patients with AAU and 39 healthy subjects (HS) underwent clinical assessment and sacroiliac joint MRI. Patients with absence of active sacroiliitis were reassessed after two years. International Spondyloarthritis Society (ASAS) classification criteria for axSpA (regardless of patient’s age) and expert opinion for definitive diagnosis of axSpA were applied. Results: Although chronic back pain was equally present in both groups, bone marrow edema (BME) in SIJ and BME highly suggestive of axSpA was found in 52 (51%) and in 33 (32%) patients with AAU compared with 11 (28%) and none in HS, respectively. Out of all AAU patients, 41 (40%) patients fulfilled the ASAS classification criteria for axSpA, and 29 (28%) patients were considered highly suggestive of axSpA based on clinical features. Two out of the 55 sacroiliitis-negative patients developed active sacroiliitis at the two-year follow-up. Conclusions: One-third of patients with AAU had active inflammation on SIJ MRI and clinical diagnosis of axSpA. Therefore, patients with AAU, especially those with chronic back pain, should be referred to a rheumatologist, and the examination should be repeated if a new feature of SpA appears.

## 1. Introduction

Non-infectious acute anterior uveitis (AAU) is an autoimmune inflammatory disease that accounts for at least 50% of the cases of non-infectious uveitis. The disease can be characterised by its genetic association to HLA-B27, which occurs in up to 50% of cases [1]. HLA-B27 is known as a highly significant risk factor for the development of spondyloarthritis (SpA), a chronic inflammatory rheumatic disease affecting the axial skeleton as well as the peripheral joints [2]. AAU is the most common extra-musculoskeletal manifestation of axial SpA (axSpA) and is found in about one-third of patients during the course of the disease and may even precede its onset [3,4,5]. AAU in patients with SpA or in the HLA-B27-positive subjects presents with specific features. It always manifests as an acute, relapsing, serofibrinous, and non-infectious intraocular inflammation unilaterally. Both eyes may be involved but usually not simultaneously. The relapses are often associated with psychological stress, excessive physical activity or virosis. After repeated attacks, uveitis may gradually progress into a chronic form in about 10% of the cases, which usually affects both eyes. Chronic AAU is associated with significantly increased sight-threatening complications such as cataract, glaucoma, and macular edema [6]. Close coordination of care between ophthalmologists and rheumatologists is necessary for the effective treatment of AAU. 

AxSpA is characterized by the presence of sacroiliitis and the development of syndesmophytes, both of which could gradually lead to the formation of sacroiliac/vertebral bony bridges and total ankylosis in some cases [7]. Sacroiliitis can be detected using several imaging techniques [8]. The most common is the conventional X-ray; however, the greatest disadvantage is its inability to capture the disease before the late stages of the inflammatory process. Alternatively, magnetic resonance imaging (MRI) is widely used for the detection of early inflammatory lesions such as bone marrow edema (BME); therefore, it has been recently used for diagnostic purposes in non-radiographic axSpA (nr-axSpA) patients, 12% of whom progress to ankylosing spondylitis (AS) over two years [9].

There are several studies on the prevalence of axSpA in patients with AAU [10,11,12]; however, the data on the presence of sacroiliitis in patients with AAU are limited. Only one study evaluated the presence of sacroiliitis using MRI; however, patients enrolled had at least two episodes of AAU [13]. Moreover, diagnostic delay in axSpA is associated with worse outcomes. Therefore, targeted screening of patients with AAU could contribute to an earlier diagnosis of axSpA. To this date, there are no official recommendations regarding whether to refer all patients with AAU for rheumatological examinations or to only examine individuals with recurrent AAU. Furthermore, repeated examination after a certain period needs to be justified.

The aim of our study was to evaluate the presence of active sacroiliitis defined by the presence of highly suggestive bone marrow edema and the prevalence of axSpA and peripheral SpA diagnosis and fulfilment of ASAS classification criteria in patients with recent, non-infectious, and serofibrinous AAU (regardless of the number of attacks) without prior rheumatologic disease, with follow-up examination after two years for patients with normal MRI findings on sacroiliac joints.

## 2. Materials and Methods

### 2.1. Patients

In total, 102 patients aged ≥18 years diagnosed with at least one episode of non-infectious AAU without prior rheumatological condition were included in the study, together with 39 healthy subjects (HS). Healthy subjects were age and sex matched, never experienced AAU, and were not treated for any rheumatological condition and inflammatory bowel disease.

The local ethics committee of the Institute of Rheumatology in Prague approved this study. Written informed consent was obtained from all individuals prior to the initiation of the study (7117/2018).

### 2.2. Clinical Assessment

Patients with uveitis of a distinct phenotype: unilateral, anterior, sudden onset (also known as acute), recurrent, serofibrinous, and non-infectious were examined, diagnosed and treated by an experienced team at the Centre for Diagnosis and Treatment of Uveitis of the General University Hospital in Prague. Patients were then referred for rheumatological assessment. The diagnosis of AAU was based on clinical examination and symptoms of the patients. Typical features of AAU included pain, photophobia, dilated ciliary vessels, inflammatory cells and precipitates on the posterior corneal surface or endothelium (keratic precipitates), and the presence of cells and flare in the anterior chamber. The cells and inflammatory proteins are sedimented at the bottom of the anterior chamber as hypopyon in the case of severe inflammation (Appendix A). The iris is swollen due to dilated vessels and the pupillary edge tends to adhere to the anterior surface of the lens, leading to posterior synechiae.

All patients with at least one AAU attack were examined no longer than three months from the onset of AAU. Participants were assessed from April 2018 to January 2021 by experienced rheumatologists; clinical assessment included history, assessment of the presence and type of back pain (chronic if lasting more than 3 months and inflammatory according to Modified Berlin criteria for inflammatory back pain), physical examination, patient-reported outcome measures and disease activity measures for axSpA: Bath Ankylosing Spondylitis Disease Activity Score (BASDAI) [14]; Ankylosing Spondylitis Disease Activity Score (ASDAS) [15]. Laboratory analysis, including the erythrocyte sedimentation rate (ESR), C-reactive protein (CRP), and HLA-B27 antigen, was performed within one week from the MRI examination (Table 1). Healthy subjects were assessed for medical history, presence and type of back pain, disease activity and laboratory tests similar to AAU patients (Table 1).

### 2.3. Imaging

MRI images were acquired using a 1.5 T scanner (Siemens). MRI examinations were conducted on both patients, with AAU as well as HS, and were performed using a standard protocol for assessing individuals with suspected axSpA, including the semicoronal short tau inversion recovery (STIR) sequence and the T1-weighted sequence (T1W) of the sacroiliac joints (SIJ) with a slice thickness of 4 mm [16].

MRI images of SIJ were evaluated for the presence of active inflammatory lesions (bone marrow edema) and structural lesions (erosion, fat metaplasia, subchondral sclerosis, ankylosis, and backfills) [17]. A trained rheumatologist blinded for clinical data first determined the presence of BME (0—absent; 1—present) and then determined whether the changes are highly suggestive (Appendix A) for axSpA (0—atypical; 1—typical) [18]. AAU patients without BME highly suggestive of axSpA were re-assessed after 2 years. Reading of all scans was performed twice within a 3-week interval. The final results from conflicting MRI scans were achieved after discussion with external musculoskeletal radiologist. 

Patients with sacroiliitis detected on MRI were further referred for X-ray examination of the SIJ and the whole spine. A trained rheumatologist and a central radiologist scored anonymized images according to the modified New York criteria for AS [19]. Any inter-observer discrepancies were resolved by secondary evaluations until an agreement was reached. 

### 2.4. Diagnosis

Patients who fulfilled the New York classification criteria [19] were classified as. Patients with no sacroiliitis on X-ray (meaning less than grade 2 bilaterally or less than grade 3 unilaterally) who had positive MRI findings (highly suggestive BME [17]) were classified according to the Assessment of SpondyloArthritis international Society (ASAS) classification criteria [20] as nr-axSpA. Both criteria were applied regardless of patient’s age due to the fact that one-fifth of the patients with AAU were over 45 years of age. Patients with peripheral involvement such as arthritis, enthesitis or dactylitis were also classified according to the ASAS classification criteria for peripheral SpA [21]. 

Patients with AAU were divided according the number of present SpA features (inflammatory back pain (IBP), alternating buttock pain, good response to non-steroid anti-inflammatory drugs, peripheral arthritis, enthesitis, dactylitis, psoriasis, inflammatory bowel disease, AAU, elevated acute phase reactants, positive family history) into three groups: highly suggestive of axSpA—if four or more SpA features (including AAU) were present regardless of HLA-B27 or if two to three SpA features (including AAU) were present together with HLA-B27; suggestive of SpA—if two to three SpA features (including AAU) were present and HLA-B27 was negative or AAU was present together with HLA-B27; non-suggestive of SpA—if AAU was present and HLA-B27 was negative or if patients did not experience chronic back pain regardless of the presence of other SpA features [22]. The diagnosis of axSpA was made by rheumatologists specialized in SpA. All sacroiliitis positive patients were referred to, followed and managed at the Early SpA Clinic of the Institute of Rheumatology in Prague. 

### 2.5. Laboratory Analysis

Fasting blood samples were collected from all individuals and were immediately centrifuged. The plasma samples were stored at −80 °C until analysis. Serum levels of CRP were determined by an immuno-turbidimetric technique using an Olympus AU 400 biochemical analyzer (Olympus Optical, Tokyo, Japan). HLA-B27 was detected by flow cytometry using an IOTest HLA-B27-FITC/HLA-B7-PE (Beckman Coulter—Immunotech SAS, Marseille, France) and a BDTM HLA-B27 Kit (BD Bioscience, San Jose, CA, USA). 

### 2.6. Statistics

Basic descriptive statistics (mean, standard deviation, median, interquartile range, skewness, and kurtosis) were computed for all variables, which were subsequently tested for normality using the Kolmogorov–Smirnov test. Differences in interval variables were tested using the Mann–Whitney U test, while the chi-square test was used to compare the frequencies of categorical variables. The bivariate relationships between the variables were assessed using the Spearman correlation coefficient. Data are presented as median (interquartile range (IQR)) if not stated otherwise. *p* values of less than 0.05 were considered statistically significant. All analyses were conducted using GraphPad Prism 5 (version 5.02; GraphPad Software, La Jolla, CA, USA).

## 3. Results

### 3.1. Patients Characteristics

The demographic and clinical characteristics of AAU patients are summarized in Table 1. Both genders in AAU patients were represented equally (52 males vs. 50 females), and the median age was 40 years. Thirty-nine HS matched by gender and age were included in the study (20 males and 19 females with a median age of 39 years, *p* = 0.815). Patients with AAU had more first-degree relatives with extra-musculoskeletal manifestation of axSpA (psoriasis, AAU, IBD) and participated less frequently in regular sport activities compared to HS (17 (17%) vs. 0 (0%) and 34 (33%) vs. 23 (59%); *p* = 0.003 and 0.007; respectively). The majority of AAU patients was HLA-B27 positive compared to HS (77 (75%) vs. 2 (5%) respectively, *p* < 0.0001). Patients with AAU had significantly higher disease activity (evaluated by BASDAI, ASDAS and CRP) compared to HS (*p* = 0.023, *p* = 0.039, and *p* = 0.014; respectively). The rest of the clinical characteristics remained comparable (Table 1). 

### 3.2. MRI of Sacroiliac Joints

BME was detected in 52 (51%) patients with AAU compared to 11 (28%) HS (*p* = 0.022). Highly suggestive BME with typical axSpA changes was detected in 33 (32%) patients with AAU. In addition, two AAU patients without BME had chronic changes highly suggestive of axSpA (erosions, fat metaplasia, sclerosis or ankylosis). There were no typical active or chronic changes in SIJ suggestive of axSpA in HS (*p* < 0.0001).

### 3.3. Differences between AAU Patients with and without MRI Defined Sacroiliitis

HLA-B27 was more frequently observed in AAU patients with sacroiliitis compared to those without (31 (89%) vs. 46 (69%) respectively, *p* = 0.030). Furthermore, more AAU patients with sacroiliitis had inflammatory back pain compared to patients without sacroiliitis (12 (34%) vs. 9 (13%) respectively, *p* = 0.020). AAU patients with sacroiliitis had significantly higher disease activity compared to patients without sacroiliitis (ASDAS: 1.46 (0.93–2.05) vs. 0.9 (0.64–1.53) respectively, *p* = 0.006; CRP: 4.43 (1.76–10.44) vs. 1.22 (0.65–2.92) respectively, *p* < 0.0001). In addition, AAU patients with sacroiliitis had worse occiput-to-wall distances compared to those without sacroiliitis (mean 1 (±2 SD) cm vs. 0 (±1 SD) cm respectively, *p* = 0.002). Other monitored parameters remained comparable (Table 2).

### 3.4. Fulfilment of the ASAS Classification Criteria for SpA (Regardless of Patient’s Age)

Out of the 102 patients with AAU, 42 patients fulfilled the ASAS classification criteria for SpA, 41 fulfilled the ASAS classification criteria for axSpA and three patients fulfilled the ASAS classification criteria for peripheral SpA (of whom two patients fulfilled both the ASAS classification criteria for axial and peripheral SpA). Furthermore, out of the 41 patients classified as axSpA, 29 patients fulfilled the imaging arm (10 patients had sacroiliitis only on MRI, and 19 patients had sacroiliitis on both MRI and X-ray), and 12 patients fulfilled the clinical arm of the ASAS classification criteria. For clarity, complete data are shown in Figure 1. Active sacroiliitis on MRI was further detected in six other patients without any back pain; therefore, they could not be classified as axSpA.

### 3.5. Clinical Diagnosis of SpA

By dividing patients according to the number of SpA features present, 29 AAU patients were indicated as highly suggestive axSpA, of whom 18 had sacroiliitis on MRI and fulfilled the imaging arm of the ASAS classification criteria for axSpA and 11 without sacroiliitis on MRI fulfilled the clinical arm of the ASAS classification criteria. Furthermore, 35 AAU patients were indicated as suggestive axSpA, of whom 11 had sacroiliitis on MRI and fulfilled the imaging arm of the ASAS classification criteria for axSpA, one without sacroiliitis on MRI fulfilled the clinical arm of the ASAS classification criteria, and 23 did not fulfil the ASAS classification criteria for axial or peripheral SpA. Thirty-six AAU patients were indicated as non-suggestive of axSpA, of whom six had sacroiliitis on MRI (and without any back pain), and none fulfilled the ASAS classification criteria for axSpA. Therefore, 100% of patients from the “highly suggestive” subset and 34% of patients from the “suggestive axSpA” subset were classified as axSpA according to the ASAS classification criteria, and 100% of non-suggestive axSpA did not fulfil the ASAS classification criteria for axSpA. For clarity, complete data are shown in Figure 2.

### 3.6. Follow-Up Examination after Two Years

Of the 67 patients with AAU and baseline MRI without highly suggestive BME, 55 patients participated in a follow-up examination after 2 years. Twelve patients did not attend this visit, mostly due to the COVID-19 pandemic. Fifty-three patients with AAU remained MRI negative (and did not fulfil the ASAS classification criteria) and two patients with chronic low back pain developed active MRI sacroiliitis. Both patients were HLA-B27 positive, the character of chronic back pain changed to inflammatory, and one patient had significantly elevated CRP levels (28.3 mg/L) on the second laboratory examination; thus, both were newly classified as axSpA.

## 4. Discussion

This is the largest study evaluating MRI of SIJ, fulfilment of the ASAS classification criteria for SpA (regardless of patient’s age), and expert clinical diagnosis of SpA in patients with AAU, including rheumatologic follow-up after two years. MRI-defined sacroiliitis (highly suggestive BME) was found in 36% of AAU patients. However, six patients were not diagnosed with axSpA due to the absence of chronic low back pain. All patients classified as highly suggestive axSpA fulfilled the ASAS classification criteria (18 fulfilled the imaging arm and 11 fulfilled the clinical arm). Follow-up examination after two years in 55 initially MRI-negative patients revealed two patients who developed axSpA with active sacroiliitis. These findings raise the question of whether all patients with AAU should undergo a targeted rheumatological examination to rule out SpA.

BME on SIJ is the first and the most typical axSpA feature which can be found on MRI. However, BME lacks sufficient level of specificity due to its manifestation in other non-rheumatological and non-inflammatory conditions [23]. Therefore, integrating “highly suggestive” BME into the criteria for axSpA has been validated [17]. We detected significantly more frequent BME in patients with AAU compared to HS, which could be caused by the proportion of patients later diagnosed with axSpA, despite the fact that a subset of HS performed more regular sports activity than patients with AAU. The presence of BME at SIJ in healthy subjects was comparable to the incidence of BME at SIJ in the non-rheumatological population [23]. Highly suggestive BME was present only in patients with AAU; however, six patients without chronic back pain could not be classified as axSpA. It is questionable whether patients with definitive sacroiliitis on imaging should be treated as axSpA to reduce the inflammatory process and to prevent further joint destructions, or the treatment should be started after the appearance of subjective signs of the disease. 

Not surprisingly, patients with AAU were more often HLA-B27 positive compared to HS, which corresponds to the well-known association between HLA-B27 and AAU, where it can be found in about 50% of all cases [1]. Patients with AAU differ from HS, especially in axSpA disease activity measures such as BASDAI, ASDAS or CRP, which were significantly worse in AAU patients, although chronic back pain and inflammatory back pain were present equally in both groups. 

Only a few studies evaluated the presence of SpA in patients with AAU [10,11,12,13]. Data on the presence of sacroiliitis are even more limited [13]. According to Sykes et al. [10], the estimated prevalence of axSpA in AAU patients is at least 20%; however, the inclusion criteria differed from ours (patients with prior SpA diagnosis were included in the analysis and 12 patients fulfilling the clinical arm of the ASAS classification criteria for axSpA were indicated as not-having axSpA by rheumatologists and were excluded). In that study, the presence of sacroiliitis on MRI of SIJ was detected in 17 patients (23%), and together with the previously mentioned 12 excluded patients, a total of 40% of examined patients without prior diagnosis of SpA fulfilled the ASAS classification criteria for axSpA, which is similar to our findings. In the study by Juanola et al., [12] the proportion of axSpA patients was 50% (of whom 60% already had the radiographic form of the disease), and peripheral SpA was 18%. The presence of sacroiliitis on MRI of SIJ was not shown. As stated by Chung et al. [11], the presence of SpA in patients with AAU was even higher (77%); however, patients were diagnosed according to The European Spondyloarthropathy Study Group (ESSG) criteria [24]. Oliviera et al. examined patients with recurrent AAU, where 40% of patients were diagnosed with axSpA. Furthermore, MRI and/or radiography-positive sacroiliitis was present in 37.5% (*n* = 9) of asymptomatic patients. Since we did not find any difference in the presence of sacroiliitis between patients with single and repeated recurrent AAU attacks, we can highlight this similarity in our findings. 

The stratification of patients according to the number of SpA features present in the highly suggestive, suggestive, and non-suggestive of axSpA subgroups has shown that all AAU patients with chronic back pain and HLA-B27 positivity should be examined by a rheumatologist, because there is strong evidence of possible axSpA diagnosis, and the additional presence of SpA features makes it more compelling. Haroon et al. [25] already proposed a possible algorithm for referral of appropriate AAU patients from ophthalmologists to rheumatologists that will aid in the early diagnosis of SpA; however, the patients with sacroiliitis without chronic back pain would be missed in this algorithm. Therefore, according to our results, even the absence of chronic back pain does not guarantee the absence of sacroiliitis; thus, it is important to search for additional SpA features and to follow up with these patients. In our study, of the 55 patients with AAU and negative sacroiliitis who were examined after two years, two patients were diagnosed as axSpA based on newly occurring inflammatory back pain, active sacroiliitis defined by MRI and abnormal CRP. Therefore, we propose that AAU patients should be referred for repeated rheumatological examinations if new symptoms of the SpA spectrum appear. 

AAU patients with sacroiliitis were significantly more often HLA-B27-positive and had worse disease activity measures compared to those without sacroiliitis, which is congruent with previous findings comparing axSpA and non-axSpA subsets [10,12]. Sykes et al. [10]. demonstrated worse BASMI scores in the subset diagnosed with axSpA, which could correspond to worse cervical spine movements in our subset of AAU patients with sacroiliitis. Furthermore, the AAU patients with sacroiliitis from our study had significantly more often IBP compared to sacroiliitis negative patients, contrary to Sykes at al. [10], where IBP remained similar between subsets.

In accordance with Juanola et al. [12], we did not find any differences in ophthalmological features of AAU between patients with and without sacroiliitis. Based on our findings, we further propose that AAU patients with chronic low back pain should be sent for rheumatological examinations regardless of AAU frequency or whether both eyes are involved.

Our study has several limitations. First, although the SIJ MRI images were evaluated by only one rheumatologist experienced in MRI readings for axSpA, MRI scans were performed twice to confirm the results. Second, the number of HS was limited. 

In conclusion, regardless of the number of attacks, patients with non-infectious AAU have an increased risk of axSpA. Our study demonstrated that one-third of patients with AAU and without prior rheumatological diagnosis had active sacroiliitis detected by MRI, and that the probability of having MRI-detected sacroiliitis increases with the number of axSpA clinical features. One-third of patients with AAU had enough clinical features to be classified as axSpA. In addition, AAU patients with MRI-defined sacroiliitis were more often HLA-B27 positive, had higher disease activity and were more likely to have inflammatory back pain compared to AAU patients without active sacroiliitis on MRI, although highly suggestive BME can also be found in some AAU patients without any back pain. It is therefore important to refer AAU patients for repeated rheumatological examinations if new symptoms from the SpA spectrum appear.

## Figures and Tables

**Figure 1 diagnostics-12-00161-f001:**
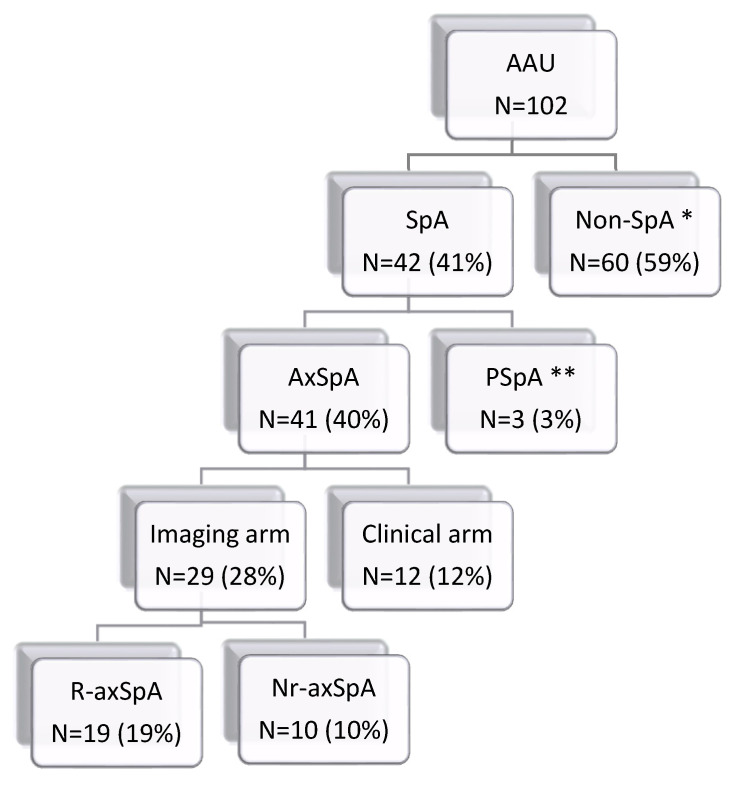
Distribution of patients with acute anterior uveitis according to ASAS classification criteria for axial and peripheral spondyloarthritis. Abbreviations: AAU, anterior uveitis patients; SpA, spondyloarthritis; AxSpA, axial spondyloarthritis; PSpA, peripheral spondyloarthritis; R-axSpA, radiographic axSpA, nr-axSpA, non-radiographic axSpA; * including six patients with sacroiliitis lacking back pain; ** two patients fulfilled both axial and peripheral axSpA.

**Figure 2 diagnostics-12-00161-f002:**
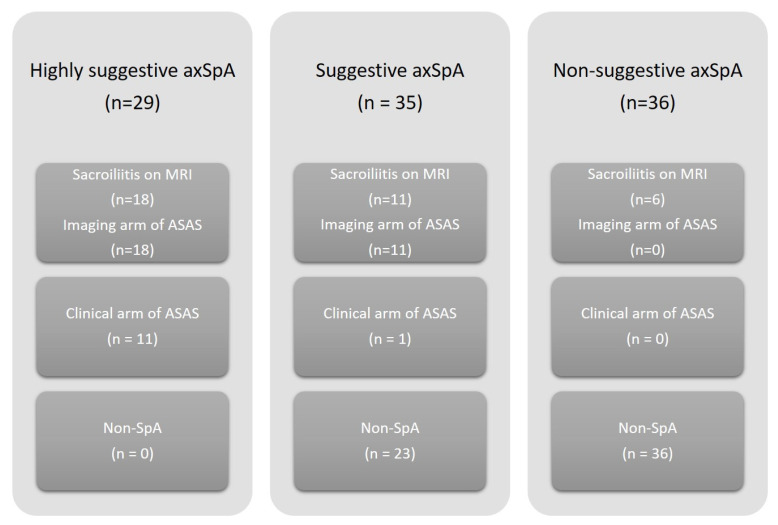
Distribution of patients with acute anterior uveitis according to present clinical SpA features and their classification according to ASAS classification criteria for axSpA. Highly suggestive of axSpA—if ≥4 SpA features were present regardless of HLA-B27 or if 2–3 SpA features were present together with HLA-B27. Suggestive of axSpA—if 2–3 SpA were present and HLA-B27 was absent or 1 SpA feature was present together with HLA-B27. Nonsuggestive of axSpA—if 1 SpA feature was present and HLA-B27 was absent or if patients did not experience chronic back pain. Abbreviations: SpA, spondyloarthritis; AxSpA, axial spondyloarthritis; ASAS, Assessment of SpondyloArthritis International Society Classification criteria for axSpA.

**Table 1 diagnostics-12-00161-t001:** Clinical characteristics of patients with anterior uveitis and healthy subjects (HS). Bold shows statistically significant values.

Characteristics	AAU Patients (*n* = 102)	HS (*n* = 39)	*p*
Gender, f (%)/m (%)	50 (49)/52 (51)	19 (49)/20 (51)	1.000
Age (years), median (IQR)	40 (32–45)	39 (34–46)	0.815
Smokers, *n* (%)	32 (31)	13 (33)	0.685
AxSpA in FDR, *n* (%)	6 (6)	0 (0)	0.187
EMS in FDR, *n* (%)	17 (17)	0 (0)	**0.003**
Play sports regularly, *n* (%)	34 (33)	23 (59)	**0.007**
Spine injury, *n* (%)	5 (5)	0 (0)	0.322
Physically demanding occupation, *n* (%)	17 (17)		
Age of AAU onset (years), median (IQR)	34 (27–42)		
AAU disease duration (years), median (IQR)	2 (0–7)		
AAU relapse (*n*), median (IQR)	2 (1–5)		
Both eyes involvement, *n* (%)	33 (32)		
HLA-B27, *n* (%)	77 (75)	2 (5)	**<0.0001**
BP, *n* (%)	73 (72)	27 (70)	0.685
IBP, *n* (%)	21 (21)	5 (13)	0.340
BASDAI, median (IQR)	1 (0.23–2.06)	0.3 (0–1.5)	**0.023**
ASDAS-CRP, median (IQR)	1.04 (0.69–1.78)	0.69 (0.64–1.22)	**0.039**
VAS (mm), median (IQR)	0 (0–20)		
CRP (mg/L), median (IQR)	1.78 (1.83–4.85)	1.13 (0.47–2.02)	**0.014**
Metrology:			
Schober test (cm), median (IQR)	5 (4–6)		
Chin-chest test (cm), median (IQR)	0 (0–0)		
Chest expansion test (cm), median (IQR)	5 (3–6)		
Occiput to wall test (cm), median (IQR)	0 (0–0)		
BME, *n* (%)	52 (51)	11 (28)	**0.022**
Sacroiliitis by X-ray *, *n* (%)	20 (57)		
Sacroiliitis by MRI **, *n* (%)	35 (34)	0 (0)	**<0.0001**

AAU, anterior uveitis; *n*, number; f, female; m, male; IQR, interquartile range; AxSpA, axial spondyloarthritis; FDR, first degree relatives; EMS, extra-musculoskeletal manifestations; BP, chronic back pain; IBP, inflammatory back pain; BASDAI, Bath Ankylosing Spondylitis Disease Activity Score; ASDAS, Ankylosing Spondylitis Disease Activity Score; VAS, visual analogue scale; CRP, C-reactive protein; BME, bone marrow edema; MRI, magnetic resonance imaging; *, New York classification criteria (1984) were applied; **, ASAS classification criteria for axSpA (2009) were applied.

**Table 2 diagnostics-12-00161-t002:** Clinical characteristics of patients with anterior uveitis and MRI defined sacroiliitis (positive) and without MRI defined sacroiliitis (negative). Bold shows statistically significant values.

Characteristics	MRI Positive (*n* = 35)	MRI Negative (*n* = 67)	*p*
Gender, f (%)/m (%)	13 (37)/22 (63)	37 (55)/30 (45)	0.098
Age (years), median (IQR)	40 (32–44)	40 (32–46)	0.657
BMI	24.2 (22.1–27.2)	24.9 (22.2–27.7)	0.640
Smokers, *n* (%)	12 (34)	20 (30)	0.659
AxSpA in FDR, *n* (%)	3 (9)	3 (5)	0.406
EMS in FDR, *n* (%)	4 (11)	13 (19)	0.406
Play sports regularly, *n* (%)	12 (34)	22 (33)	0.659
Spine injury, *n* (%)	2 (6)	3 (5)	>0.9999
Physically demanding occupation, *n* (%)	9 (26)	8 (12)	0.096
Age of AAU onset (years), median (IQR)	33 (26–40)	34 (28–44)	0.182
AAU disease duration (years), median (IQR)	2 (0–8)	1 (0–6)	0.334
AAU relapse (*n*), median (IQR)	2 (1–5)	2 (1–4)	0.787
Both eyes involvement, *n* (%)	15 (43)	18 (27)	0.121
HLA-B27, *n* (%)	31 (89)	46 (69)	**0.030**
BP, *n* (%)	29 (83)	44 (66)	0.105
IBP, *n* (%)	12 (34)	9 (13)	**0.020**
BASDAI, median (IQR)	1.1 (0.25–2.26)	1 (0.2–1.9)	0.732
ASDAS-CRP, median (IQR)	1.46 (0.93–2.05)	0.9 (0.64–1.53)	**0.006**
VAS (mm), median (IQR)	0 (0–12)	0 (0–20)	0.480
CRP (mg/L), median (IQR)	4.43 (1.76–10.44)	1.22 (0.65–2.92)	**<0.0001**
Metrology:			
Schober test (cm), mean (±SD)	4.8 (±1.3)	5 (±1.2)	0.669
Chin-chest test (cm), mean (±SD)	0.4 (±1.0)	0.4 (±1.0)	0.530
Chest expansion test (cm), mean (±SD)	4.1 (±2.0)	4.9 (±2.0)	0.090
Occiput to wall test (cm), mean (±SD)	0.8 (±1.9)	0.1 (±0.8)	**0.002**
BME, *n* (%)	33 (94)	19 (28)	**<0.0001**

AAU, anterior uveitis; *n*, number; f, female; m, male; IQR, interquartile range; BMI, body mass index; AxSpA, axial spondyloarthritis; FDR, first degree relatives; EMSAM, extra-musculoskeletal manifestations; BP, chronic back pain; IBP, inflammatory back pain; BASDAI, Bath Ankylosing Spondylitis Disease Activity Score; ASDAS, Ankylosing Spondylitis Disease Activity Score; VAS, Visual analogue scale; CRP, C-reactive protein; BME, bone marrow edema.

## Data Availability

Data are available for all on request.

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
