# Peer review of "The Prevalence of MRI-Defined Sacroiliitis and Classification of Spondyloarthritis in Patients with Acute Anterior Uveitis: A Longitudinal Single-Centre Cohort Study"

_diagnostics, 2022, doi:10.3390/diagnostics12010161_

Round 1

Reviewer 1 Report

The study has interesting practical implications. However, there are some issues to address:

  1. In abstract: "ASAS classification criteria for axSpA for definitive diagnosis of axSpA were applied". The ASAS classification criteria are NOT diagnostic criteria. The goal of the criteria is to create a homogenous population of patients with an already established expert diagnosis of SpA for research purposes. As far as I'm concerned, you can apply the criteria in a non-SpA population to address the lack of specificity, but you can't use them to diagnose a patient with SpA. This is very important, as these criteria are often wrongfully used as diagnostic criteria in routine clinical care. Please make sure that the criteria are not used to make an SpA diagnosis.
  2. Methods: please list in- and exclusion criteria for healthy subjects and if they were age-matched. 
  3. Methods: "Out of 67 AAU patients without BME..." Figures should be in the results section. Just mention that subjects without BME were re-assessed after two years. 
  4. Methods: "Reading of all scans was performed twice..." How were conflicting results handled?
  5. Methods, section 2.4 Diagnosis: suggestive of SpA and non-suggestive of SpA -> change to axSpA, as patients with several SpA features including peripheral arthritis but no back pain, can be very suggestive of pSpA. 
  6. Results: how many had structural lesions and which structural lesions were present? Prior research has shown that structural lesions seem more specific for axSpA compared to BME. This was not addressed in this paper. 
  7. Results, 3.2 MRI of SIJ: "...was detected in 33 patients with AAU" -> list percentage. 
  8. Everywhere back pain is mentioned: always chronic back pain (entry criterion for classification criteria!), if yes, please add "chronic". In addition, was age taken into account for the classification criteria (<45 years old)? 
  9. Results, 3.4: "Furthermore, 29 out of all...." -> repetition from a couple of sentences above. Please delete. 
  10. Discussion: "However, six patients simultaneously met the ASAS classification criteria for..." I don't understand this sentence. Where is it in the results? And how can one fulfill the classification criteria for axSpA without having chronic back pain as it is an entry-criterion?
  11. "Even the absence of chronic back pain does not guarantee the absence of sacroiliitis." That's true, but what is the clinical relevance of this sacroiliitis? Prior research has shown that non-SpA subjects often display sacroiliitis suggestive of axSpA... This may mean nothing and indicates the lack of specificity of MRI...

Author Response

Response to Reviewer 1

Comment 1:

In abstract: "ASAS classification criteria for axSpA for definitive diagnosis of axSpA were applied". The ASAS classification criteria are NOT diagnostic criteria. The goal of the criteria is to create a homogenous population of patients with an already established expert diagnosis of SpA for research purposes. As far as I'm concerned, you can apply the criteria in a non-SpA population to address the lack of specificity, but you can't use them to diagnose a patient with SpA. This is very important, as these criteria are often wrongfully used as diagnostic criteria in routine clinical care. Please make sure that the criteria are not used to make a SpA diagnosis.

Answer 1:

Thank you for the comment. We are aware of the fact that ASAS classification criteria for axSpA are not supposed to be used for diagnostic purposes. We have not used ASAS classification criteria to make a SpA diagnosis. However, since diagnostic criteria for axSpA are lacking, we decided to use these criteria to show readers the portion of patients who fulfil ASAS classification criteria, including radiographic arm and clinical arm , as well as expert opinion for the definitive diagnosis of axSpA.

We decided to emphasize it also in the abstract as follows:

… International Spondyloarthritis Society (ASAS) classification criteria for axSpA and expert opinion for definitive diagnosis of axSpA were applied…

Comment 2: Methods: please list in- and exclusion criteria for healthy subjects and if they were age-matched. 

Answer 2: Thank you for the comment. Inclusion and exclusion criteria for healthy subjects were added to the text as follows:

… Healthy subjects were age and sex matched, have not ever experienced acute anterior uveitis, were not treated for any rheumatological condition and inflammatory bowel disease…

Comment 3: Methods: "Out of 67 AAU patients without BME..." Figures should be in the results section. Just mention that subjects without BME were re-assessed after two years. 

Answer 3: Thank you for the comment. Text was changed as follows:

…AAU patients without BME highly suggestive of axSpA were re-assessed after 2 years...

Comment 4: Methods: "Reading of all scans was performed twice..." How were conflicting results handled?

Answer 4: Thank you for the comment. The final results from conflicting MRI scans were achieved after discussion with external musculoskeletal radiologist. (which was added to the text).

Comment 5: Methods, section 2.4 Diagnosis: suggestive of SpA and non-suggestive of SpA -> change to axSpA, as patients with several SpA features including peripheral arthritis but no back pain, can be very suggestive of pSpA. 

Answer 5: Thank you for the comment. We changed SpA to axSpA.

Comment 6: Results: how many had structural lesions and which structural lesions were present? Prior research has shown that structural lesions seem more specific for axSpA compared to BME. This was not addressed in this paper.

Answer 6: Thank you for the comment. The focus of this paper was primarily on active inflammatory changes (bone marrow edema). We did not assess chronic inflammatory lesions in patients with present BME but in those without BME where chronic changes indicating past sacroiliitis (homogenous and sharply defined fat metaplasia together with erosions) were present in two patients.

Comment 7: Results, 3.2 MRI of SIJ: "...was detected in 33 patients with AAU" -> list percentage. 

Answer 7: Thank you for the comment. Text was changed as follows: … was detected in 33 (32%) patients…

Comment 8: Everywhere back pain is mentioned: always chronic back pain (entry criterion for classification criteria!), if yes, please add "chronic". In addition, was age taken into account for the classification criteria (<45 years old)? 

Answer 8: Thank you for the comment. By back pain we meant chronic back pain, therefore „chronic“ was added in the text together with its criteria in the Methods section. Furthermore, we did not mention in the text that we applied inflammatory back pain together with ASAS classification criteria for axSpA regardless of patients age. We have corrected our mistake and emphasized it severally in the text.

…Both criteria were applied regardless of patients age due to the fact that one fifth of the patients with AAU was over 45 years…

Comment 9: Results, 3.4: "Furthermore, 29 out of all...." -> repetition from a couple of sentences above. Please delete. 

Answer 9: Thank you for the comment. The text was deleted.

Comment 10: Discussion: "However, six patients simultaneously met the ASAS classification criteria for..." I don't understand this sentence. Where is it in the results? And how can one fulfill the classification criteria for axSpA without having chronic back pain as it is an entry-criterion?

Answer 10: Thank you for the comment. This sentence was misunderstood by a native English speaker. We corrected it as follows:

However, six patients were not diagnosed with axSpA due to the absence of chronic low back pain.

Comment 11: "Even the absence of chronic back pain does not guarantee the absence of sacroiliitis." That's true, but what is the clinical relevance of this sacroiliitis? Prior research has shown that non-SpA subjects often display sacroiliitis suggestive of axSpA... This may mean nothing and indicates the lack of specificity of MRI...

Answer 11: Thank you for the comment. We think it is clinically relevant. In our opinion, follow-up of the patient without back pain with an image of active sacroiliitis by rheumatologists is important because of increased risk of axSpA clinical onset.

Reviewer 2 Report

ABSTRACT

  • Suggest replacing “common” manifestation of axSpA with “relatively common”.
  • Out of all: Do you mean “Out of all patients with AAU”?

INTRODUCTION

  • Please replace “extraarticular” with “extra-musculoskeletal” manifestations because it’s the currently accepted nomenclature (in the introduction and throughout the whole manuscript).

AIM

  • Please specify “MRI” sacroiliitis
  • Confirm if you mean axSpA and pSpA “diagnosis” or “classification criteria”

METHODS

  • Explain how the healthy controls were recruited, and if they were screening for any back/ joint pain. Indicate here that they matched with the patients for age and gender.
  • Indicate in the exclusion criteria whether patients with previously known SpA were excluded from this study.

RESULTS

  • TABLE 1: Sacroiliitis by x-ray / by MRI: explain which criteria were used to define (New York? ASAS?)
  • TABLE 2: How is the difference in Occiput to wall test statistically significant although its value = 0 in both MRI positive and negative participants?

  • FIGURE S2: Suggest to show the T1 version of this MRI slice, if available.

Author Response

Response to Reviewer 2

Comment 1: ABSTRACT

  • Suggest replacing “common” manifestation of axSpA with “relatively common”.
  • Out of all: Do you mean “Out of all patients with AAU”?

Answer 1:

Thank you for the comments. We replaced the text for „relatively common“ and further changed the text as follows:

…Out of all patients with AAU…

 Comment 2: INTRODUCTION

  • Please replace “extraarticular” with “extra-musculoskeletal” manifestations because it’s the currently accepted nomenclature (in the introduction and throughout the whole manuscript).

AIM

  • Please specify “MRI” sacroiliitis
  • Confirm if you mean axSpA and pSpA “diagnosis” or “classification criteria”

Answer 2:

Thank you for the comments.

The „extra-musculoskeletal“ was used in the text.

Aims were corrected as follows:

…The aim of our study was to evaluate the presence of active sacroiliitis defined by presence of highly suggestive bone marrow edema and the prevalence of axSpA and peripheral SpA diagnosis and fulfilment of ASAS classification criteria in patients with recent, non-infectious, and serofibrinous AAU…

Comment 3: METHODS

  • Explain how the healthy controls were recruited, and if they were screening for any back/ joint pain. Indicate here that they matched with the patients for age and gender.
  • Indicate in the exclusion criteria whether patients with previously known SpA were excluded from this study.

Answer 3: Thank you for the comments. The text was extended as follows:

…102 patients aged ≥ 18 years diagnosed with at least one episode of non-infectious AAU without prior rheumatological condition were included in the study, together with 39 healthy subjects (HS). Healthy subjects were age and sex matched, have not ever experienced acute anterior uveitis, were not treated for any rheumatological condition and inflammatory bowel disease.

… Healthy subjects were assessed for medical history, for presence and type of back pain, disease activity and laboratory tests similarly to AAU patients (Table 1).

Comment 4: RESULTS

  • TABLE 1: Sacroiliitis by x-ray / by MRI: explain which criteria were used to define (New York? ASAS?)
  • TABLE 2: How is the difference in Occiput to wall test statistically significant although its value = 0 in both MRI positive and negative participants?
  • FIGURE S2: Suggest to show the T1 version of this MRI slice, if available.

Answer 4: Thank you for the comments. New abbreviations were added in Table 1. The significant difference in occiput to the wall test is better seen if mean with SD are applied (0.8 (±1.9) vs. 0.1 (±0.8) cm), therefore we replaced it in the Table 2. The T1 sequence of MRI was added to S2.

Round 2

Reviewer 1 Report

The authors addressed the comments and questions sufficiently.